# Robust and local quantum self-testing of systems of arbitrary high dimension

Kishor Bharti,[1, 2, *] Maharshi Ray,[2, *] Antonios Varvitsiotis,[3] Adán Cabello,[4, 5] and Leong-Chuan Kwek[2, 6, 7]

[1]*Institute of High Performance Computing, Agency for Science,*
*Technology and Research (A\*STAR), 1 Fusionopolis Way, Connexis, Singapore 138632, Singapore*
[2]*Centre for Quantum Technologies, National University of Singapore*
[3]*Engineering Systems and Design Pillar, Singapore University of Technology and Design*
[4]*Departamento de Física Aplicada II, Universidad de Sevilla, E-41012 Sevilla, Spain*
[5]*Instituto Carlos I de Física Teórica y Computacional, Universidad de Sevilla, E-41012 Sevilla, Spain*
[6]*MajuLab, CNRS-UNS-NUS-NTU International Joint Research Unit, Singapore UMI 3654, Singapore*
[7]*National Institute of Education, Nanyang Technological University, Singapore 637616, Singapore*

We address the problem of certifying quantum systems of arbitrary dimension (e.g., quantum computers) without making assumptions about their inner working (i.e., in a device-independent way) and without assuming that they are entangled with other systems (thus excluding self-testing methods based on violations of Bell inequalities). We show that there is a family of noncontextuality inequalities which can be expressed as a sum of probabilities of events whose graph of exclusivity is an odd antihole (i.e., the complement of a cycle with an odd number $n$ of vertices larger than three) whose maximum quantum violation is only achieved with quantum systems of dimension $d = n - 2$ and is self-testable (i.e., unique up to isometries). We show that the elements of this family can be used for robust self-testing of quantum states and measurements of systems of arbitrarily high odd dimension in experiments with sequential measurements. An extra assumption needed is that measurements are ideal (i.e., minimally disturbing), which can be falsified by observation. In addition, we introduce a protocol to determine whether the quantum state and measurements maximally violating any non-contextuality inequality which can be expressed as a sum of probabilities of events is provably non-self-testable.

## I. INTRODUCTION

Classical computing devices store and manipulate sequences of binary numbers to perform computational tasks that are relevant to the society such as healthcare scheduling, weather prediction, and routing of vehicles. However, as the number of variables involved in a computational task grows beyond a limit, such as in the simulation of $50 - 60$ spin-half particles, even the state-of-the-art supercomputers fail spectacularly to carry out the required computations. Harnessing the properties of quantum theory to carry out computations beyond the reach of classical supercomputers is one major objective of quantum information processing. Quantum algorithms choreograph the state of a number of qubits (the quantum analogue of classical bits) in an intelligent fashion, allowing us to perform highly complex computations. Nevertheless, the ability of a quantum device to carry out a specified set of instructions relies crucially on this ability to faithfully generate on demand specific quantum states and perform specific measurements on them.

We expect future quantum resources to be shared through cloud or other internet accesses. In a faithfully minimalist scenario, developing trust in the functionality of quantum devices necessitates certification schemes that do not rely on any assumptions of their inner workings. One of the most important approaches for establishing trust in third-party quantum hardware is through the notion of self-testing [1–6]. In this setting,

the quantum device is modeled as a black box and interactions with the device correspond to measurement operations. Typical self-testing results correspond to guarantees regarding the uniqueness of the measurement settings and the underlying state preparation, based solely on the measurement statistics. The majority of existing self-testing results rely on the use of Bell inequalities, i.e., linear expressions involving probabilities of measurement-outcome events that can be carried out in a Bell experiment [7], whereas more recent works extend the self-testing paradigm beyond the scope of Bell scenarios [8, 9].

Self-testing based on Bell experiments is a powerful technique that allows us to develop insights concerning the inner workings of a pair of non-communicating entangled devices, i.e., to verify distributed quantum hardware. However, as computing typically takes place in a localized setting, any scheme for certifying the functionality of a programmable quantum computer must be of a local nature to be of any practical relevance. The first such local self-testing scheme was introduced in [10] within the framework of contextuality, which constitutes a broad generalization of Bell non-locality [7, 11]. However, the contextuality-based self-testing scheme from [10] was limited in scope due to its applicability to three dimensional quantum systems. We also note the recent work [12] which investigates the possibility of a self-testing-like certification of a single untrusted quantum device. They show how a classical verifier can robustly certify that a single computationally bounded quantum prover must have prepared an EPR pair. Our main result in this article is a self-testing scheme for

* equal contribution

local certification of programmable quantum devices of arbitrary high dimensionality.

A contextuality scenario is characterized by a set of measurement events, where two events are mutually exclusive if they correspond to the same measurement but different outcomes. The exclusivity relations between the measurement events can be conveniently encoded as edges in an undirected graph, called the exclusivity graph. The seemingly simple idea to use graphs to represent exclusivity relations has spearheaded the development of a new line of research at the interface of graph theory and contextuality [13]. The research on contextuality has led to many foundational and practical results [14–20]. The linear inequalities, the violation of which witnesses contextuality, are referred to as noncontextuality inequalities. Bell inequalities are a special type of noncontextuality inequalities, where the "contexts" are provided via the space-like separation of the parties involved [7, 21]. The first "local" noncontextuality inequality violated by quantum theory was identified by Klyachko, Can, Binicioğlu, and Shumovsky (KCBS) [22]. The bound on a noncontextuality inequality in a noncontextual hidden variable (NCHV) model is called the NCHV bound. Quantum theory violates the NCHV bound for suitably chosen state and measurement settings, and thus manifests as a contextual theory. To any exclusivity graph, we associate a "canonical" noncontextuality inequality. Furthermore, we say that a graph is self-testable, if the corresponding noncontextuality inequality admits self-testing.

## II.  ASSUMPTIONS AND RESULTS

*Main results—* The main contributions of this work can be summarized as follows:

1. We introduce a local and robust self-testing scheme for certifying high-dimensional programmable quantum devices based on contextuality. As a key ingredient in the scheme, we show that the family of odd anti-cyclic graphs with at least five vertices are self-testable. This allows for self-testing pure states and projective measurements of arbitrarily high odd dimension.

2. We present a protocol to determine whether a given graph is provably non-self-testable (see Appendix IV). This protocol goes beyond the results in Ref. [10] which only provide sufficient conditions for self-testing. We use the protocol to prove that not all graphs with a positive gap between NCHV bound and the maximum quantum bound for the corresponding canonical noncontextuality inequality admit self-testing by providing an explicit counterexample.

*Assumptions—* Our local certification protocol is built on four key assumptions:

1. The quantum device is programmable and is error-corrected.

2. The measurements are ideal [23, 24], i.e.,

   (a) They are outcome-replicable (give the same outcome when repeated on the same physical system).

   (b) They do not disturb the compatible measurements.

   (c) Their coarse-grainings also satisfy the above two properties.

3. The measurements satisfy the compatibility structure according to an odd cycle graph.

4. The device has bounded memory.

We continue with some important comments on these four assumptions. In quantum theory, ideal measurements correspond to projective measurements and the assumption that measurements are projective can be tested experimentally [25]. Some prominent examples of experiments based on ideal measurements are [26] and [27]. Moreover, any compatibility structure corresponding to a set of ideal measurements can be tested experimentally based on outcome statistics [26]. The assumption of memory is more crucial since classical simulation of the outcome statistics corresponding to contextuality experiments is possible with memory [28–32]. As every programmable device has a finite memory, the amount of memory needed to simulate the outcome statistics will get beyond the capacity of any such classical device after a certain number of rounds of the contextuality test. Determining the exact memory cost of simulating quantum contextuality corresponding to arbitrary exclusivity graphs and building an efficient (in terms of number of rounds) protocol to mitigate the memory assumption is a promising future direction. Lastly, it is important to note that our scheme is not fully device-independent because of the above assumptions. Nevertheless, a single device cannot be self-tested in a device-independent manner, so our work gives the best possible result in this setting. Our scheme is semi-device independent with experimentally testable assumptions.

## III.  SELF-TESTING VIA ODD ANTI-CYCLES

*Graph approach to contextuality—* An arbitrary experimental scenario can be characterized by a set of measurement events $e_1, \ldots, e_n$. Two events are mutually exclusive if they correspond to same measurement but different outcomes. The exclusivity structure of a set of measurement events is captured by the exclusivity graph, denoted $\mathcal{G}_{ex}$, with nodes $\{1, \ldots, n\}$ (denoted by $[n]$) corresponding to events $\{e_i\}_{i=1}^n$. Two nodes $i$ and $j$ are adjacent (denoted by $i \sim j$) if $e_i$ and $e_j$ are mutually exclusive.

Given an experimental scenario with an exclusivity graph $\mathcal{G}_{\mathrm{ex}}$, a theory assigns probability to the events corresponding to its vertices. The mapping $p : [n] \to [0,1]$, where $p_i + p_j \leq 1$, for all $i \sim j$ is called a behavior. Here, the non-negative numbers $p_i$ refer to the probability of the event $e_i$. We call a behavior $p$ deterministic noncontextual if all the probabilities $p_i$ are either 0 or 1 and the occurrence of a event does not depend on the possibility of occurrence of other events. The convex hull of all deterministic noncontextual behaviors forms a polytope, denoted by $\mathcal{P}_{nc}(\mathcal{G}_{\mathrm{ex}})$, which contains all possible noncontextual behaviors for the experimental scenario encoded by the graph $\mathcal{G}_{\mathrm{ex}}$. The behaviors which lie outside $\mathcal{P}_{nc}(\mathcal{G}_{\mathrm{ex}})$ are contextual behaviors. The set of noncontextual behaviors is bounded by finitely many half spaces, which are called noncontextuality inequalities. Formally speaking, noncontextuality inequalities correspond to linear inequalities of the form

$$\sum_{i \in [n]} w_i p_i \leq B_{nc}(\mathcal{G}_{\mathrm{ex}}, w), \ \forall p \in \mathcal{P}_{nc}(\mathcal{G}_{\mathrm{ex}}), \qquad (1)$$

where $w_1, \ldots, w_n \geq 0$ and $B_{nc}(\mathcal{G}_{\mathrm{ex}}, w)$ are real scalars. The noncontextuality inequalities with all the weights $\{w_i\mathrm{s}\}$ equal to one will be referred to as canonical noncontextuality inequalities. By definition, $B_{nc}(\mathcal{G}_{\mathrm{ex}}, w)$ corresponds to the NCHV bound on the linear expression $\sum_{i \in [n]} w_i p_i$ and is also equal to the weighted independence number of the exclusivity graph $\mathcal{G}_{\mathrm{ex}}$, defined as the cardinality of the largest set of pairwise non-adjacent nodes of $\mathcal{G}_{\mathrm{ex}}$ [13]. A quantum behavior has the following form:

$$p_i = \mathrm{tr}(\rho \Pi_i), \forall i \in [n] \text{ and } \mathrm{tr}(\Pi_i \Pi_j) = 0, \text{ for } i \sim j, \quad (2)$$

for some quantum state $\rho$ and quantum projectors $\Pi_1, \ldots, \Pi_n$ acting on a Hilbert space $\mathcal{H}$. An ensemble $\rho, \{\Pi\}_{i=1}^n$ satisfying (2) is called a *quantum realisation* of the behavior $p$. For a given quantum behavior $p$, there can be multiple quantum realisations. The set of quantum behaviors is a convex set, which we denote by $\mathcal{P}_q(\mathcal{G}_{\mathrm{ex}})$. The maximum value of the linear expression $\sum_{i \in [n]} w_i p_i$, as $p$ ranges over the set of quantum behaviors, $\mathcal{P}_q(\mathcal{G}_{\mathrm{ex}})$ can exceed the classical bound. We will denote the maximum attainable quantum value by $B_{cq}(\mathcal{G}_{\mathrm{ex}}, w)$. Interestingly, $B_{cq}(\mathcal{G}_{\mathrm{ex}}, w)$ is equal to the *Lovász theta number* of the graph $\mathcal{G}_{\mathrm{ex}}$ and admits a formulation as a tractable optimisation problem known as a semidefinite program [13] (see Methods Section).

*Robust self-testing—* Informally speaking, a noncontextuality inequality $\mathcal{I}$ is said to self-test a quantum realisation $\rho, \{\Pi\}_{i=1}^n$ if it achieves the quantum bound for the noncontextuality inequality $\mathcal{I}$ and furthermore, all other quantum realisations which achieve the quantum bound corresponding to $\mathcal{I}$ are equivalent to $\rho, \{\Pi\}_{i=1}^n$ up to global isometry. For a formal definition, the reader is referred to

Section B in the Appendix. As discussed in [10], the essential ingredient in proving self-testing results for a noncontextuality inequality $\mathcal{I}$ with underlying exclusivity graph $\mathcal{G}_{\mathrm{ex}}$ is that the corresponding Lovász theta semidefinite program (cf. ($P_G$)) has an unique optimal solution. In the case of the KCBS inequality, the exclusivity graph is a pentagon. The configuration corresponding to optimal quantum violation admits an umbrella structure. The KCBS inequality has been generalized to odd $n$-cycle exclusivity graphs, which are called $\mathrm{KCBS_n}$ inequalities. The $\mathrm{KCBS_n}$ inequalities are the canonical nonconcontextuality inequalities for an odd cycle graph and admit robust self-testing [10].

*Self-testing anti-cycles—* Building on the link between graph theory and contextuality [33], in combination with the strong perfect graph theorem [34], it was shown in [13, 33] that the presence of certain exclusivity structures is a necessary and sufficient condition for a noncontextuality scenario to witness quantum contextuality. These fundamental exclusivity structures correspond to an odd number of events that are either cyclically or anti-cyclically exclusive. In the graph theory literature, odd cycles and odd anti-cycles are called odd holes and odd antiholes respectively. The canonical noncontextuality inequality for an antihole is given by:

$$\sum_{i=1}^n p_i \leq 2, \text{ for all } p \in \mathcal{P}_{nc}\left(\overline{C_n}\right), \qquad (3)$$

which we call antihole inequalities and correspond to facets of the classical polytope for $\overline{C_n}$. The quantum bound for the antihole inequalities, i.e., the Lovász theta number for $\overline{C_n}$ is $\frac{1 + \cos \frac{\pi}{n}}{\cos \frac{\pi}{n}}$. A canonical quantum ensemble which achieves the quantum value for the antihole inequalities corresponding to odd $n$ is given by $n - 2$ dimensional quantum state and projectors. In fact *the dimension of quantum system achieving optimal quantum violation of antihole inequalities must be $n - 2$ for odd $n$.* We provide a proof of the aforementioned claim in Appendix D. Moreover, we show that *the antihole inequalities admit robust self-testing.* This fact is proven in the Appendix (Sections B and C).

As the presence of holes and/or antiholes in a contextuality scenario dictates the possibility of quantum advantage [33], our result regarding antihole self-testing in combination with [10] imply that *all noncontextuality inequalities which are fundamental to quantum theory admit local robust self-testing.* This is because the generalized KCBS and antihole inequalities are the unique facet-defining noncontextuality inequalities for their respective odd hole and antihole exclusivity scenarios [35].

## IV. NOT ALL NONCONTEXTUALITY INEQUALITIES ADMIT SELF-TESTING

We have proved that all fundamental noncontextuality inequalities admit self-testing. A natural question is whether every noncontextuality inequality with separation between corresponding noncontextual hidden variable bound and quantum bound admits self-testing. Below we provide an explicit noncontextuality inequality which shows that the answer is negative. In graph theoretical terms, we identify a non-perfect graph whose Lovász theta SDP admits multiple primal optimal solutions. We make crucial use of the following result [36, Theorem 5] to determine the (non)uniqueness of primal optimal under strict complementarity, see also [37].

**Theorem 1.** *Let $(X^*, Z^*)$ be a pair of primal and dual optimal solutions satisfying strict complementarity. Then, uniqueness of $X^*$ implies that $Z^*$ is dual nondegenerate.*

The exclusivity graph of our counter-example is shown in Figure 1. The corresponding canonical noncontextuality inequality is given by

$$\sum_{i=1}^{6} p_i \leq 2, \tag{4}$$

whose quantum bound is equal to $\sqrt{5}$.

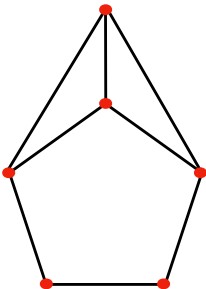

FIG. 1. The above exclusivity graph corresponds to the canonical noncontextuality inequality with minimal number of measurement events which doesn't admit self-testing.

Consider the pair of primal-dual optimal solutions

$$Z^* = \begin{bmatrix} \sqrt{5} & -1 & -1 & -1 & -1 & -1 & -1 \\ \hline -1 & 1 & 0 & c & c & 0 & c \\ -1 & 0 & 1 & 0 & c & c & c \\ -1 & c & 0 & 1 & 0 & c & 0 \\ -1 & c & c & 0 & 1 & 0 & 1 \\ -1 & 0 & c & c & 0 & 1 & 0 \\ -1 & c & c & 0 & 1 & 0 & 1 \end{bmatrix}, \tag{5}$$

where $c = \frac{\sqrt{5}-1}{2}$ and

$$X^* = \begin{bmatrix} 1 & f & f & f & h & f & h \\ \hline f & f & k & 0 & 0 & k & 0 \\ f & k & f & k & 0 & 0 & 0 \\ f & 0 & k & f & r & 0 & r \\ h & 0 & 0 & r & h & r & 0 \\ f & k & 0 & 0 & r & f & r \\ h & 0 & 0 & r & 0 & r & h \end{bmatrix}, \tag{6}$$

where $f = \frac{1}{\sqrt{5}}$, $h = \frac{f}{2}$, $k = \frac{5-\sqrt{5}}{10}$ and $r = \frac{k}{2}$. Since rank$(Z^*) = 3$ and rank$(X)^* = 4$, strict complementarity holds. Using Theorem 1, the uniqueness of $X^*$ implies dual nondegeneracy. To determine dual nondegeneracy for $Z^*$ we (once again) resort to solving a system of linear equations. The symmetric variable matrix $M$ is given by

$$M = \begin{bmatrix} 0 & m_0 & m_1 & m_2 & m_3 & m_4 & m_5 \\ \hline m_0 & m_0 & m_6 & 0 & 0 & m_7 & 0 \\ m_1 & m_6 & m_1 & m_8 & 0 & 0 & 0 \\ m_2 & 0 & m_8 & m_2 & m_9 & 0 & m_{10} \\ m_3 & 0 & 0 & m_9 & m_3 & m_{11} & 0 \\ m_4 & m_7 & 0 & 0 & m_{11} & m_4 & m_{12} \\ m_5 & 0 & 0 & m_{10} & 0 & m_{12} & m_5 \end{bmatrix}, \tag{7}$$

Solving for the linear systems of equations $Z^* M = 0$, we get $m_{11} = -m_{12}$, $m_{10} = m_{12}$, $m_5 = \frac{1+\sqrt{5}}{2} m_{12}$, $m_9 = -m_{12}$, $m_3 = -\left(\frac{1+\sqrt{5}}{2}\right) m_{12}$, $m_0 = m_1 = m_2 = m_4 = m_6 = m_7 = m_8 = 0$. For example, if we set $m_{12} = 1$, we can get a consistent assignment of $m_i$, from $i = 0$ to 12, which isn't all zero. Hence, the dual solution $Z^*$ is degenerate, which together with strict complementarity implies that the primal is not unique. Thus the noncontextuality inequality in (4) does not admit self-testing.

We also report that we found several other non-perfect graphs ( equivalently noncontextuality inequalities) which do not admit self-testing. Identifying the exact classes of graphs which admit self-testing will be interesting but we leave that as an open question.

## V. TOOLS AND TECHNIQUES

The main tool we use in this work to show that antihole inequalities admit robust self-testing is Theorem 2, shown in [10], which provides a sufficient condition for a graph to be self-testable. This result relies crucially on the rich properties of a powerful class of mathematical optimisation models, known as Semidefinite programs (SDPs) (see Appendix A). SDPs constitute a vast generalisation of linear optimisation models where scalar variables are replaced by vectors and the constraints and objective function are affine in terms of the inner products of the vectors. Equivalently, collecting all pairwise inner products of these vectors in matrix, known as the Gram matrix, an SDP corresponds to optimising a linear function of the Gram matrix subject

to affine constraints. Analogously to linear programs, to any SDP there is an associated a dual program whose value is equal to the primal under reasonable assumptions. Next, we single out certain properties of primal-dual solutions that are of relevance to his work. A pair of primal dual optimal solutions $(X^*, Z^*)$ with no duality gap (i.e., $\text{tr}(X^* Z^*) = 0$), satisfies *strict complementarity* if the $\text{Range}(X^*)$ and $\text{Range}(Z^*)$ give a direct sum decomposition of the underlying space. Furthermore, an optimal dual solution $Z^*$ with rank $r$ is *dual nondegenerate* if the tangent space at $Z^*$ of the manifold of symmetric matrices with rank equal to $r$ together with the linear space of matrices defining the SDP span the entire space of symmetric matrices. In this work we focus on the Lovász theta SDP, see ($P_G$) in Appendix B. The proof of our main result involves two main steps. First, we construct a dual optimal solution of ($D_G$) for an odd-cycle graph by providing an explicit mapping between the Gram vectors of a primal ($P_G$) optimal solution of an odd-cycle graph and the Gram vectors of a dual optimal solution of the complement graph; see Theorem 3 for details. Next, once we construct a dual optimal solution, we show in Theorem 4 that it satisfies the non-degeneracy conditions given in (B4). By Theorem 2 , this shows that antihole inequalities admit robust self-testing. Details of the proofs can be found in Appendix C. Additionally, we show that not all graphs admit self-testing by providing a counter example of such a graph (see Appendix IV). The overall scheme for determining whether a graph is self-testable (equivalently whether the primal optimal solution of the Lovász theta SDP corresponding to that graph is unique or not) is provided in the form of a flowchart in Figure 2.

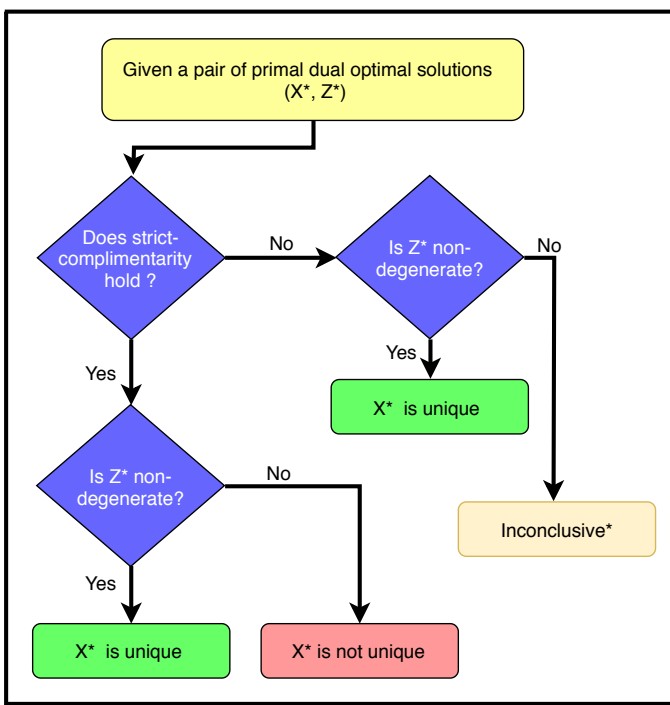

FIG. 2. Flowchart for determining (non)uniqueness of primal solution(s). Here inconclusive* refers to the fact that one can still hope to arrive at a definitive answer by restarting the algorithm using a different dual optimal solution (if it exists).

## VI. OPEN PROBLEMS

We proceed with some mathematical questions of physical relevance that are raised by our work. First, the dimension of the quantum system corresponding to the maximum quantum violation of a noncontextuality inequality is given by the rank of the corresponding primal optimal matrix of the semidefinite program. In this work we proved that the dimension of the quantum system must be at least $n - 2$, which corresponds to the maximum violation of the anti-cycle inequality with $n$ vertices in its exclusivity graph. To make our analysis robust with respect to experimental errors, it is important to identify lower bounds on the dimension of the quantum system when we tolerate $\epsilon$-suboptimal solutions.

Second, the outcome statistics corresponding to contextuality experiments can be classically simulated using memory. As discussed above, one way to mitigate this is to run the contextuality tests for sufficiently large number of rounds so that the programmable device runs out of memory. One future direction will be to determine the exact memory cost of simulating quantum contextuality corresponding to arbitrary exclusivity graphs. Estimating the memory cost of simulating contextuality is comparatively easier for state-independent scenarios. It will be interesting to develop self-testing schemes for measurement settings corresponding to state-independent noncontextuality inequalities. Another exciting direction would be to explore if it's possible to prevent the prover from classically simulating the output statistics using cryptographic tools, such as blind quantum computing [38]. This will also ensure that the protocol is efficient in terms of the number of testing rounds.

Third, the noise robustness of a noncontextuality inequality can be quantified by the ratio of the corresponding quantum bound $\vartheta(\mathcal{G}_{\text{ex}}, w)$ to the classical bound $\alpha(\mathcal{G}_{\text{ex}}, w)$. For anti-cycle inequalities, the ratio tends to one, and thus the self-testing scheme is not noise-tolerant. However, there are graphs for which the quantum to classical ratio scales with the number of vertices of the underlying exclusivity graph [39]. It will be exciting to provide self-testing schemes for such noise-tolerant graphs, as this could be used to self test NISQ devices.

## VII. ACKNOWLEDGEMENTS

We thank the National Research Foundation of Singapore, the Ministry of Education of Singapore, MINECO Project No. FIS2017-89609-P with FEDER funds, and the Knut and Alice Wallenberg Foundation for financial support. A significant part of the research project Local certification of programmable quantum devices of arbitrary high dimensionalitywas carried out during "New directions in quantum information" conference organized by Nordita, the Nordic Institute for Theoretical Physics.

## APPENDIX

### Appendix A: Semidefinite programming basics

A semidefinite program (SDP) is given by an optimisation problem of the following form

$$\sup_X \left\{ \langle C, X \rangle : X \in \mathcal{S}_+^n, \ \langle A_i, X \rangle = b_i \ (i \in [m]) \right\}, \quad \text{(P)}$$

where $\mathcal{S}_+^n$ denotes the cone of $n \times n$ real positive semidefinite matrices and $\langle X, Y \rangle = \text{tr}(X^T Y)$. The corresponding dual problem is given by

$$\inf_{y,Z} \left\{ \sum_{i=1}^m b_i y_i \ : \ \sum_{i=1}^m y_i A_i - C = Z \in \mathcal{S}_+^n \right\}. \quad \text{(D)}$$

A pair of primal-dual optimal solutions $(X^*, Z^*)$ with no duality gap (i.e., $\text{tr}(X^* Z^*) = 0$), satisfies *strict complementarity* if

$$\text{rank}(X^*) + \text{rank}(Z^*) = n. \quad \text{(A1)}$$

Lastly, an optimal dual solution $Z^*$ is called *dual nondegenerate* if the linear system in the symmetric matrix variable $M$

$$MZ^* = \text{tr}(MA_1) = \ldots = \text{tr}(MA_m) = 0, \quad \text{(A2)}$$

only admits the trivial solution $M = 0$.

Central to this work is the Lovász theta SDP corresponding to a graph $G$, whose primal formulation is:

$$\begin{aligned}
\vartheta(G) = \max \ & \sum_{i=1}^n X_{ii} \\
\text{s.t. } & X_{ii} = X_{0i}, \ i \in [n], \\
& X_{ij} = 0, \ i \sim j, \\
& X_{00} = 1, \ X \in \mathcal{S}_+^{1+n},
\end{aligned} \quad \text{(}P_G\text{)}$$

and the dual formulation we use is given by:

$$\begin{aligned}
\vartheta(G) = \min \ & Z_{00} \\
\text{s.t. } & Z_{ii} = -(2Z_{0i} + 1), \ i \in [n], \\
& Z_{ij} = 0, \ i \not\sim j, \\
& Z \in \mathcal{S}_+^{1+n}.
\end{aligned} \quad \text{(}D_G\text{)}$$

### Appendix B: Robust self-testing

To prove our main result we use the following definitions from [10]. A noncontextuality inequality $\sum_{i \in [n]} w_i p_i \leq B_{nc}(\mathcal{G}_{\text{ex}}, w)$ is a *self-test* for the realisation $\{|u_i\rangle\langle u_i|\}_{i=0}^n$ if:

1. $\{|u_i\rangle\langle u_i|\}_{i=0}^n$ achieves the quantum supremum $B_{qc}(\mathcal{G}_{\text{ex}}, w)$;

2. For any other realisation $\{|u_i'\rangle\langle u_i'|\}_{i=0}^n$ that also achieves $B_{qc}(\mathcal{G}_{\text{ex}}, w)$, there exists an isometry $V$ such that

$$V|u_i\rangle\langle u_i|V^\dagger = |u_i'\rangle\langle u_i'|, \quad 0 \leq i \leq n. \quad \text{(B1)}$$

Furthermore, a noncontextuality inequality $\sum_{i \in [n]} w_i p_i \leq B_{nc}(\mathcal{G}_{\text{ex}}, w)$ is an $(\epsilon, r)$-robust self-test for $\{|u_i\rangle\langle u_i|\}_{i=0}^n$ if it is a self-test, and furthermore, for any other realisation $\{|u_i'\rangle\langle u_i'|\}_{i=0}^n$ satisfying

$$\sum_{i=1}^n w_i |\langle u_i'|u_0'\rangle|^2 \geq B_{qc}(\mathcal{G}_{\text{ex}}, w) - \epsilon,$$

there exists an isometry $V$ such that

$$\||V|u_i\rangle\langle u_i|V^\dagger - |u_i'\rangle\langle u_i'|\| \leq \mathcal{O}(\epsilon^r), \quad 0 \leq i \leq n. \quad \text{(B2)}$$

The proof of our main result hinges on the following theorem (first introduced in [10]):

**Theorem 2.** *Consider a noncontextuality inequality $\sum_{i=1}^n w_i p_i \leq B_{nc}(\mathcal{G}_{\text{ex}}, w)$. Assume that*

1. *There exists an optimal quantum realisation $\{|u_i\rangle\langle u_i|\}_{i=0}^n$ such that*

$$\sum_i w_i |\langle u_i|u_0\rangle|^2 = B_{qc}(\mathcal{G}_{\text{ex}}, w) \quad \text{(B3)}$$

*and $\langle u_0|u_i\rangle \neq 0$, for all $1 \leq i \leq n$, and*

2. *There exists a dual optimal solution $Z^*$ for the SDP $(D_G)$ such that the homogeneous linear system*

$$\begin{aligned}
& M_{0i} = M_{ii}, \ \text{for all } 1 \leq i \leq n, \\
& M_{ij} = 0, \ \text{for all } i \sim j, \\
& MZ^* = 0,
\end{aligned} \quad \text{(B4)}$$

*in the symmetric matrix variable $M$ only admits the trivial solution $M = 0$.*

*Then, the noncontextuality inequality is an $(\epsilon, \frac{1}{2})$-robust self-test for $\{|u_i\rangle\langle u_i|\}_{i=0}^n$.*

### Appendix C: Self-testing antihole inequalities

The antihole noncontextuality inequalities are given by $\sum_{i=1}^n p_i \leq 2$ for all $p \in \mathcal{P}_{nc}(\overline{C_n})$. The quantum

bound for the antihole inequalities, i.e., the Lovász theta number for $\overline{C_n}$ is $\frac{1+\cos\frac{\pi}{n}}{\cos\frac{\pi}{n}}$ [40]. A canonical quantum ensemble which achieves the quantum value for the antihole inequalities corresponding to odd $n$ is given by $n-2$ dimensional quantum state and projectors. Explicitly, the quantum state is

$$|v_0\rangle = (1,0,\ldots,0)^T, \tag{C1}$$

but the description of the projectors $\{\Pi_j = |v_j\rangle\langle v_j|\}_{j=1}^n$ is more involved [33]. Let us denote the $k$-th component of $|v_j\rangle$ corresponding to projector $\Pi_j$ as $v_{j,k}$. For $0 \le j \le n-1$ and $0 \le k \le n-3$,

$$v_{j,0} = \sqrt{\frac{\vartheta(\overline{C_n})}{n}} \tag{C2}$$

$$v_{j,2m-1} = T_{j,m}\cos R_{j,m} \tag{C3}$$

$$v_{j,2m} = T_{j,m}\sin R_{j,m} \tag{C4}$$

for $m = 1, 2, \ldots \frac{n-3}{2}$ and

$$T_{j,m} = (-1)^{j(m+1)}\sqrt{\frac{2\cos(\frac{\pi}{n}) + (-1)^{m+1}\cos\left(\frac{(m+1)\pi}{n}\right)}{n\cos\frac{\pi}{n}}}, \tag{C5}$$

$$R_{j,m} = \frac{j(m+1)\pi}{n}. \tag{C6}$$

For the antihole noncontextuality inequalities, the ensemble described above achieves the quantum value and satisfies the first condition of Theorem 2. It remains to establish the existence of a dual optimal solution for the SDP corresponding to the Lovász theta number of antihole graphs such that the conditions in (B4) are satisfied. Towards this goal, we first proceed to provide the explicit form of the dual optimal solution.

**Theorem 3.** *Let $X^* = \mathrm{Gram}(v_0, v_1, \ldots, v_n)$ be the unique optimal solution for $(P_{C_n})$. Then,*

$$Z_n^* = \vartheta(\overline{C_n})\mathrm{Gram}(-v_0, v_1, \ldots, v_n) \tag{C7}$$

*is a dual optimal solution for $(D_{\overline{C_n}})$. Another useful expression for $Z_n^*$ is given by*

$$Z_n^\star = \left[\begin{array}{c|c} \vartheta(\overline{C_n}) & -e^\top \\ \hline -e & circ(u) \end{array}\right] \in \mathbb{R}^{(1+n)\times(1+n)}, \tag{C8}$$

*where $e$ is the vector of all ones of length $n$,*

$$u = (1, \vartheta(\overline{C_n})\langle v_1|v_2\rangle, \ldots, \vartheta(\overline{C_n})\langle v_1|v_n\rangle), \tag{C9}$$

*and $circ(\cdot)$ maps an $n$-dimensional vector and outputs the corresponding circulant matrix.*

*Proof.* It was shown in [10] that $(P_{C_n})$ admits a unique optimal solution $X^*$. For any $k = 0, 1, \ldots, n-1$, the map taking $i \to i+1$ (modulo $n$) is an automorphism of $C_n$ (i.e., a bijective map that preserves adjacency and non-adjacency). In particular, this implies that $X^*$ is circulant and furthermore, constant along each band. Specifically, all diagonal entries of $X^*$ are equal, and as $\vartheta(C_n) = \sum_{i=1} X_{ii}^*$, it follows that

$$\langle v_i|v_i\rangle = X_{ii}^* = \vartheta(C_n)/n. \tag{C10}$$

Analogously, for a pair of indices $i, j$ with $|i-j| = k$ we have that $X_{ij}^* = \langle v_1|v_{k+1}\rangle$. Moreover, the feasibility of $X^*$ we have that $X_{00}^* = \langle v_0|v_0\rangle = 1$. Thus $Z_{00}^* = \vartheta(\overline{C_n})$ has the correct value and it remains to show that $Z^*$ is feasible. Next, by feasibility of $X^*$ we have that $X_{ij}^* = \langle v_i|v_j\rangle = 0$, when $i \sim j$ in $C_n$. Thus, by definition of $Z^*$ we have that $Z_{ij}^* = \vartheta(\overline{C_n})\langle v_i|v_j\rangle = 0$ for all edges of $C_n$. Finally we show that $Z_{ii}^* = -(2Z_{0i}^* + 1)$, $i \in [n]$. Indeed,

$$Z_{ii}^* = \langle v_i|v_i\rangle\vartheta(\overline{C_n}) = \frac{\vartheta(C_n)}{n}\vartheta(\overline{C_n}) = 1, \tag{C11}$$

where we used (C10) and that $\vartheta(C_n)\vartheta(\overline{C_n}) = n$ (see Theorem 8 of [41]). To finish the proof we note that

$$\begin{aligned} -(2Z_{0i}^* + 1) &= -(2\vartheta(\overline{C_n})\langle -v_0|v_i\rangle + 1) \\ &= 2\vartheta(\overline{C_n})\langle v_0|v_i\rangle - 1 \\ &= 2\vartheta(\overline{C_n})\langle v_i|v_i\rangle - 1 \\ &= 1. \end{aligned} \tag{C12}$$

where the second last equality follows from the constraint that $\langle v_0|v_i\rangle = \langle v_i|v_i\rangle$ for $i \in [n]$ and the last equality follows by substituting $\langle v_i|v_i\rangle\vartheta(\overline{C_n}) = 1$. $\square$

Finally, we show that the dual optimal solution satisfies the conditions in B4.

**Theorem 4.** *The dual optimal solution $Z^*$ corresponding to the complement of an odd-cycle graph satisfies the conditions in B4.*

*Proof.* We show that for any odd $n$, the only symmetric matrix $M \in \mathbb{R}^{(1+n)\times(1+n)}$ satisfying

$$M_{00} = 0, \ M_{0i} = M_{ii}, \ M_{ij} = 0 \ (\forall\, i \sim j), \ MZ^* = 0, \tag{C13}$$

is the matrix $M = 0$, where $i \sim j$ here refers to an edge in the $\overline{C_n}$ graph. Barring the $MZ^* = 0$ constraint, the rest already guarantee that there are at most $2n$ potentially non-zero entries in the $M$ matrix (not counting the repeated entries) corresponding to $\overline{C_n}$ graph. Let the first row of $M$ be $(0, m_1, m_2, \ldots, m_n)$. We fill the rest of the potential non-zero slots in $M$ with $m_{n+1}, m_{n+2}, \ldots, m_{2n}$.

For example, for $n = 7$, we have

$$M_7 = \begin{pmatrix} 0 & m_1 & m_2 & m_3 & m_4 & m_5 & m_6 & m_7 \\ \hline m_1 & m_1 & m_8 & 0 & 0 & 0 & 0 & m_{14} \\ m_2 & m_8 & m_2 & m_9 & 0 & 0 & 0 & 0 \\ m_3 & 0 & m_9 & m_3 & m_{10} & 0 & 0 & 0 \\ m_4 & 0 & 0 & m_{10} & m_4 & m_{11} & 0 & 0 \\ m_5 & 0 & 0 & 0 & m_{11} & m_5 & m_{12} & 0 \\ m_6 & 0 & 0 & 0 & 0 & m_{12} & m_6 & m_{13} \\ m_7 & m_{14} & 0 & 0 & 0 & 0 & m_{13} & m_7 \end{pmatrix},$$
(C14)

For notational convenience, let $MZ^* = \begin{pmatrix} q & r^\top \\ r & T \end{pmatrix}$, where $T \in \mathbb{R}^{n \times n}$, $q \in \mathbb{R}$ and $r \in \mathbb{R}^{n \times 1}$. In the rest of this section we use the notation $\underline{i}$ to denote $i \bmod n$, where $i$ is an integer. The linear equation corresponding to $q$ implies that

$$\sum_{i=1}^n m_i = 0. \tag{C15}$$

For $i \in \{1, 2, \ldots, n\}$, the $2n$ linear equations corresponding to $T_{i,\underline{i+1}}$ and $T_{i,\underline{i-1}}$ imply

$$m_{n+1} = m_{n+2} = \ldots = m_{2n}. \tag{C16}$$

Now consider the $n$ linear equations corresponding to $r$. These equations along with C16 imply

$$m_1 = m_2 = \ldots = m_n. \tag{C17}$$

Using C17 along with C15 we have

$$m_1 = m_2 = \ldots = m_n = 0. \tag{C18}$$

Finally, using the equations corresponding to $r$ again and C18 we have

$$m_{n+1} = m_{n+2} = \ldots = m_{2n} = 0, \tag{C19}$$

implying that $M = 0$, as desired. $\qquad\square$

### Appendix D: Dimension for optimal violation of antihole inequalities

**Theorem 5.** *Given an antihole noncontextuality inequality with an odd number of $n$ measurement events, the quantum system achieving the optimal quantum bound must be at least $(n-2)$ dimensional.*

*Proof.* The value of a noncontextuality inequality achievable within quantum theory is equal to the weighted Lovász theta number of the underlying graph $G$ and admits the SDP formulation ($P_G$). The lower bound on the dimension of a quantum system achieving the optimal quantum bound is the rank of the unique primal optimal matrix

$$X_n^\star = \left[ \begin{array}{c|c} \frac{1}{} & \frac{\vartheta(\overline{C_n})}{n} e^\top \\ \hline \frac{\vartheta(\overline{C_n})}{n} e & \mathrm{circ}(u) \end{array} \right], \tag{D1}$$

where $e$ is the all-ones vector of length $n$, $\mathrm{circ}(\cdot)$ is the circulant function that takes as input a $n$ dimension vector and outputs a $n \times n$ matrix with the input vector as its top row and every subsequent row being one place right shifted modulo $n$ and $u = (\frac{\vartheta(\overline{C_n})}{n}, \frac{n-\vartheta(C_n)}{2\vartheta(C_n)^2}, 0, 0, 0, \ldots, 0, 0, \frac{n-\vartheta(C_n)}{2\vartheta(C_n)^2})$. Since $X_n^\star$ is real, its rank over complex field is the same as over real field and equals to the number of nonzero eigenvalues (with multiplicity). Furthermore, a lower bound on the rank of $X_n^\star$ is given by the rank of the lower right block matrix (the circulant portion). The eigenvalues of a circulant matrix can be calculated easily using the circulant vector. A few lines of algebra yields the following expression for the eigenvalues of the lower right block matrix,

$$\lambda_j = \frac{1}{\vartheta_n} + \frac{n - \vartheta_n}{\vartheta_n} \cos\left(\frac{2\pi j}{n}\right) \tag{D2}$$

for $j \in [n]$ and $\vartheta_n$ denotes the Lovász theta number for the holes with odd $n$. One can see that $\lambda_j \neq 0$ unless $j = \frac{n-1}{2}$ or $\frac{n+1}{2}$. Thus, the rank of the circulant matrix is $n-2$ for all odd values of $n$. Thus, the lower bound on the rank of the optimal feasible matrix $X^\star$ is $n-2$ which is same as the lower bound on the dimension of the desired quantum system. $\qquad\square$

### Appendix E: Complex versus Real SDPs

**Lemma 6.** *Consider a real SDP*

$$\sup_X \left\{ \langle C, X \rangle : X \in \mathcal{S}_+^n, \ \langle A_i, X \rangle = b_i \ (i \in [m]) \right\}, \tag{E1}$$

*that admits a unique optimal solution $X^*$ witnessed by a dual nondegenerate optimal solution $Z^*$. Then, the SDP considered over the complex numbers, i.e.,*

$$\sup_X \left\{ \langle C, X \rangle_{\mathbb{C}} : X \in \mathcal{H}_+^n, \ \langle A_i, X \rangle_{\mathbb{C}} = b_i \ (i \in [m]) \right\}, \tag{$P_{\mathbb{C}}$}$$

*still admits a unique optimal solution, where $\langle X, Y \rangle_{\mathbb{C}} = \mathrm{tr}(X^\dagger Y)$ and $\mathcal{H}_+^n$ denotes the set of $n \times n$ Hermitian positive semidefinite matrices.*

*Proof.* First, we show that the study of a complex SDP can be reduced to an equivalent real SDP. This fact is well known but we provide a brief argument for completeness. Indeed, for any feasible solution $X = X_{\mathbb{R}} + iX_{\mathbb{C}} \in \mathcal{H}_+^n$, the constraint $\langle A_i, X \rangle_{\mathbb{C}} = b_i$ is equivalent to two constraints on its real and imaginary part, namely: $\langle A_i, X_{\mathbb{R}} \rangle = b_i$ and $\langle A_i, X_{\mathbb{C}} \rangle = 0$. Furthermore, checking whether $X_{\mathbb{R}} + iX_{\mathbb{C}}$ is Hermitian PSD is equivalent to

$$\begin{pmatrix} X_{\mathbb{R}} & -X_{\mathbb{C}} \\ X_{\mathbb{C}} & X_{\mathbb{R}} \end{pmatrix} \in \mathcal{S}_+^n. \tag{E2}$$

Based on these observations we define the realification of

$(P_\mathbb{C})$ as the following SDP over the real numbers:

$$\sup_{X,Y} \langle C, X \rangle$$
$$\text{s.t. } \langle A_i, X \rangle = b_i \ (i \in [m])$$
$$\langle A_i, Y \rangle = 0 \ (i \in [m]) \qquad (P_\mathbb{R})$$
$$\begin{pmatrix} X & -Y \\ Y & X \end{pmatrix} \in \mathcal{S}_+^{2n}.$$

Clearly, the solutions of $(P_\mathbb{C})$ are in bijection with the solutions of the realification, and thus, to show that $(P_\mathbb{C})$ has a unique solution it suffices to show that $(P_\mathbb{R})$ has a unique solution. Bringing $(P_\mathbb{R})$ into standard SDP form we arrive at the formulation:

$$\sup_W \begin{pmatrix} C/2 & 0 \\ 0 & C/2 \end{pmatrix} \bullet W$$
$$\text{s.t. } \begin{pmatrix} A_i/2 & 0 \\ 0 & A_i/2 \end{pmatrix} \bullet W = b_i \ (i \in [m])$$
$$\begin{pmatrix} 0 & A_i/2 \\ A_i/2 & 0 \end{pmatrix} \bullet W = 0 \ (i \in [m]) \qquad (E3)$$
$$X = Z, \quad Y + Y^T = 0,$$
$$W = \begin{pmatrix} X & Y \\ Y^T & Z \end{pmatrix} \in \mathcal{S}_+^{2n},$$

whose dual is to minimize the function $\sum_{i=1}^m \lambda_i b_i$ over all $\lambda_i, \mu_j, t_{ij}, z_{ij}$ satisfying

$$\sum_{i=1}^m \lambda_i \begin{pmatrix} A_i/2 & 0 \\ 0 & A_i/2 \end{pmatrix} + \sum_{i=1}^m \mu_i \begin{pmatrix} 0 & A_i/2 \\ A_i/2 & 0 \end{pmatrix} +$$
$$\sum_{ij=1}^n t_{ij} \begin{pmatrix} E_{ij} & 0 \\ 0 & -E_{ij} \end{pmatrix} + \sum_{ij=1}^n z_{ij} \begin{pmatrix} 0 & E_{ij} \\ E_{ij}^T & 0 \end{pmatrix} - \begin{pmatrix} C/2 & 0 \\ 0 & C/2 \end{pmatrix} \succeq 0.$$
$$(E4)$$

We conclude the proof by showing that $\begin{pmatrix} Y^* & 0 \\ 0 & Y^* \end{pmatrix}$ is a dual nondegenerate optimal solution for the realification. First, by dual feasibility we have $Y^* = \sum_{i=1}^m y_i^* A_i - C$ for appropriate scalars $y_i^*$. Setting $\lambda_i = y_i^*$ and all other dual variables to zero, we have established feasibility. Second, to show optimality note that $\begin{pmatrix} X^* & 0 \\ 0 & 0 \end{pmatrix}$ is optimal for the realification, and furthermore, $\begin{pmatrix} X^* & 0 \\ 0 & 0 \end{pmatrix} \bullet \begin{pmatrix} Y^* & 0 \\ 0 & Y^* \end{pmatrix} = 0$. Lastly, to check nondegeneracy consider a symmetric matrix $M = \begin{pmatrix} M_1 & M_2 \\ M_2^T & M_3 \end{pmatrix}$ satisfying

$$0 = \begin{pmatrix} M_1 & M_2 \\ M_2^T & M_3 \end{pmatrix} \begin{pmatrix} Y^* & 0 \\ 0 & Y^* \end{pmatrix} \qquad (E5)$$

and

$$0 = M \bullet \begin{pmatrix} A_i/2 & 0 \\ 0 & A_i/2 \end{pmatrix} = M \bullet \begin{pmatrix} 0 & A_i/2 \\ A_i/2 & 0 \end{pmatrix} =$$
$$= M \bullet \begin{pmatrix} E_{ij} & 0 \\ 0 & -E_{ij} \end{pmatrix} = M \bullet \begin{pmatrix} 0 & E_{ij} \\ E_{ij}^T & 0 \end{pmatrix}. \qquad (E6)$$
$$M_1 Y^* = M_2 Y^* = M_3 Y^* = 0. \qquad (E7)$$

Furthermore, using (E6), from the third equation we get $M_1 = M_3$, from the first one we get $\langle M_1, A_i \rangle = 0$, and from the second one $\langle M_2, A_i \rangle = 0$. Summarizing, for all $k = 1, 2, 3$ we have that

$$M_k Z^* = 0 \text{ and } \langle M_k, A_i \rangle = 0 \ (i \in [m]).$$

As $Z^*$ is dual nondegenerate, it has the property that, for any $M \in \mathcal{S}^n$,

$$MZ^* = \langle M, A_i \rangle = 0 \ \forall i \implies M = 0.$$

Putting everything together, we get $M_1 = M_2 = M_3 = 0$. $\square$

---

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
