# Peer review of "Robust and local quantum self-testing of systems of arbitrary high dimension"

_SciPost Physics_

## Round 1 · Referee Report · Anonymous (Referee 1) · 2023-3-25

Report

The paper under review presents a derivation of local self-testing schemes for quantum systems of arbitrary high odd dimension using noncontextuality inequalities based on the graph representation of exclusivity relationships. The authors' approach seems theoretically sound, and their results on self-testing in contextuality scenarios are interesting.

However, I have concerns regarding the authors' framing of their results as a "certification scheme for programmable quantum devices." The certification scheme proposed in the paper is of a very different nature than certification schemes for other quantum devices such as QKD and RNG. Even if we assume that ideal measurements can be done, which is never practically achievable, the certification only confirms that the device is capable of preparing the self-tested states and performing the required measurements. This is not what a user would be interested in when buying access to a cloud-based quantum computer. It does not guarantee that the computing company has not built a specialized device that can prepare the specific states required for self-testing.

To address this concern, if the authors insist on presenting this as primarily a way to 'certify programmable devices', the authors should honestly discuss the limitations of their proposed certification scheme and its relevance to real-world applications. More reasonably, they could present their results on self-testing in contextuality scenarios as the main focus of the paper and discuss the potential implications for certification in the discussion section.

---

## Round 1 · Referee Report · Anonymous (Referee 2) · 2023-5-30

Report

Designing methods that allow the end-user to certify or verify whether a quantum device operates in agreement with its specification is certainly a vital problem in the field of quantum technologies. This paper proposes a possible solution to this problem which draws from the field of quantum contextuality. As it is argued in the paper, the latter is well suited for certification of devices which are localized in a single place, as opposed to Bell nonlocality which can be used for certification of distributed devices.

The authors prove that a certain family of noncontextuality inequalities corresponding to anti-cyclic graphs with odd number n of vertices (called also antihole inequalities) have a unique quantum maximiser in Hilbert spaces of dimension n-2 up to (global) unitary transformations. (The optimal state and rank-one projective measurements are provided in the paper.) The authors also prove this statement to be robust to noises and experimental imperfections. This result is a basis for the certification method. The classical verifier programs the device to perform the contextuality experiment that leads to maximal violation of one of these inequalities, and, if the maximal violation is observed, then the verifier is sure that the device operates as expected. The method rellies on a few assumptions for which the authors provide some justification.

(*)Another result that this work delivers is an example of a noncontextuality inequality, which despite being nontrivial, does not admit self-testing in the sense of not having a unique maximizer in a given dimension.

All these results sound nice and might be of interest to people working on quantum certification as well as on quantum contextuality. I have, however, some reservations that do not allow me to give the final recommendation already now. First of all, the idea of using nonclassical effects of quantum theory such as Bell nonlocality or contextuality for certification of quantum devices is not that new and has been already discussed in the literature. This is not a serious problem. What concerns me more is how the results are phrased. For instance, the main body of the manuscript offers a proof of self-testing for the antihole inequalities, however, this self-testing is proven according to a quite restrictive definition which assumes that the measurements are not only projective, which due to Naimark dilation is ultimately fine, but also that the measurement operators are rank-one. This assumption constrains the dimension of the underlying Hilbert space to the number of measurements plus one (corresponding to the state) and limits the possibility of finding other optimal higher-dimensional quantum realizations that might not be equivalent to those found in the paper.

I would therefore suggest that the authors slightly rephrase their statements; in particular I suggest to move the definition of self-testing they use from the appendices to the main body of the paper and clearly state the underlying assumptions. It would also be desirable to confront the present definition with that proposed by Irfan et al., Phys. Rev. A 101, 032106 (2020), which appears to be less restrictive. Perhaps it would also make sense to replace ‘self-testing’ by ‘semi-device-independent certification’; the notion ‘self-testing’ was put forward in the device-independent setting, while the results here rely on many additional assumptions.

Below I provide some further suggestions to be taken into account while preparing a new version of the manuscript.

Comments:

  1. In the abstract the authors say We address the problem of certifying quantum systems of arbitrary dimension … (i.e., in a device-independent way), whereas later, at the end of Sec. II they say ‘Our scheme is semi-device-independent …’. The abstract thus suggests that the results are stronger than they really are.

  2. The authors might want to update some of the references such as for instance [9].

  3. When explaining the original notion of self-testing the Authors might want to give credit to the work of Mayers and Yao.

  4. Concerning the result (*) above, it is already known that there are Bell inequalities that despite being nontrivial are not self-tests because they are maximally violated by various pure states or inequivalent measurements choices. Bell inequalities are particular examples of noncontextuality inequalities. The authors might want to mention that in the paper.

  5. The definition of self-testing provided in the text slightly differs from that stated in Appendix B. Precisely, the one in the main body seems to allow for arbitrary projective measurements, whereas the one in Appendix B is formulated only for rank-one projective measurements. This is another motivation to move the definition of self-testing the authors really use to the main body.

  6. On page 3 the authors refer to Methods section, which doesn’t exist. I guess the authors meant Sec. V (‘Tools and techniques’).

  7. I don’t understand why it is stated in the paper that the dimension of the quantum system achieving optimal quantum violation … must be n-2 for odd n. I can always tensor the n-2 dim representation of the measurements with the identity of arbitrary size and the optimal state with some arbitrary state and obtain another optimal quantum realization of higher dimension. Theorem 5 in Appendix D correctly provides this information, saying that the dimension must be at least n-2. Again, the statements in the main body slightly depart from what is proven in the appendices.

---

## Editorial Decision

awaiting_resubmission